# On the Spatial Distribution of Temporal Complexity in Resting State and Task Functional MRI

**DOI:** 10.3390/e24081148

**Published:** 2022-08-18

**Authors:** Amir Omidvarnia, Raphaël Liégeois, Enrico Amico, Maria Giulia Preti, Andrew Zalesky, Dimitri Van De Ville

**Affiliations:** 1Applied Machine Learning Group, Institute of Neuroscience and Medicine, Forschungszentrum Juelich, 52428 Juelich, Germany; 2Institute of Systems Neuroscience, Heinrich Heine University Duesseldorf, 40225 Duesseldorf, Germany; 3Neuro-X Institute, École Polytechnique Fédérale de Lausanne, 1202 Geneva, Switzerland; 4Department of Radiology and Medical Informatics, University of Geneva, 1211 Geneva, Switzerland; 5CIBM Center for Biomedical Imaging, 1015 Lausanne, Switzerland; 6Melbourne Neuropsychiatry Centre, Department of Psychiatry, The University of Melbourne, Melbourne, VIC 3010, Australia; 7Department of Biomedical Engineering, The University of Melbourne, Melbourne, VIC 3010, Australia

**Keywords:** functional MRI, resting state, task engagement, temporal complexity, multiscale entropy, Hurst exponent, task specificity, graph signal processing

## Abstract

Measuring the temporal complexity of functional MRI (fMRI) time series is one approach to assess how brain activity changes over time. In fact, hemodynamic response of the brain is known to exhibit critical behaviour at the edge between order and disorder. In this study, we aimed to revisit the spatial distribution of temporal complexity in resting state and task fMRI of 100 unrelated subjects from the Human Connectome Project (HCP). First, we compared two common choices of complexity measures, i.e., Hurst exponent and multiscale entropy, and observed a high spatial similarity between them. Second, we considered four tasks in the HCP dataset (Language, Motor, Social, and Working Memory) and found high task-specific complexity, even when the task design was regressed out. For the significance thresholding of brain complexity maps, we used a statistical framework based on graph signal processing that incorporates the structural connectome to develop the null distributions of fMRI complexity. The results suggest that the frontoparietal, dorsal attention, visual, and default mode networks represent stronger complex behaviour than the rest of the brain, irrespective of the task engagement. In sum, the findings support the hypothesis of fMRI temporal complexity as a marker of cognition.

## 1. Introduction

The concept of complexity has been studied in many real-world phenomena including mechanical systems [1,2], volcanic eruption [3], climate change [4], earthquakes [5], financial markets [6], biological signals [7,8,9,10,11,12], and the hemodynamics of the human brain [13,14,15,16,17]. Fluctuations of spontaneous brain activity, measured by resting state functional magnetic resonance imaging (rsfMRI), are of particular importance because they can provide insight into brain structures and their functional relationships [18]. The analysis of fMRI time series has revealed distinguishable functional communities across brain areas called resting state networks (RSNs) [19], which have also been reported during task engagement [20,21]. Within and between-RSN fMRI signals have been associated with changes in the dynamical states of brain function [22,23,24].

RSNs exhibit a balanced dynamic between order and disorder in time and space, referred to as *spatiotemporal complexity*. This phenomenon is linked with brain anatomy and has been studied through whole-brain modelling and chaos theory [25,26], as well as a combination of information theory with network science [27,28]. Along the time axis, this feature of brain function is reduced to *temporal complexity* and has been shown to be reproducible and distinguishable from head motion [14]. It seems also RSN-specific and more pronounced in high-level functional networks such as the default mode network (DMN), frontoparietal network (FP), and dorsal attention network (DA) [13,14]. The complex dynamic of brain function originates from a large number of interacting components in the cerebral cortex that are often divided into several subunits themselves with distinctive functional properties. The collective activity of these modules leads to a non-centralized and self-organized behaviour with diverse realizations in the time domain [29].

Logarithmic linearity in the frequency domain is one of the manifestations of temporal complexity that has been reported in fMRI. Ciuciu et al. [21] showed that fMRI signals have scale-free and multifractal properties during rest and task performance. Scale-free dynamic of fMRI is likely affected by mental states. In fact, task engagement may suppress self-similarity of fMRI [30]. McDonough and Nashiro [13] hypothesized that RSNs may present characteristic complexity patterns. The region-specific properties of fMRI were also studied in [31] where higher irregularity was reported in sub-cortical regions such as the caudate, the olfactory gyrus, the amygdala, and the hippocampus, whilst primary sensorimotor and visual areas were associated with slower temporal changes. Nezafati et al. [22] confirmed this finding and also, showed that networks exhibit distinct complex properties which may change between the resting state and during task performance. Omidvarnia et al. [14] reproduced the findings of RSN-specific temporal complexity in rsfMRI and the lower complexity of sub-cortical regions in contrast to cortical networks across 987 healthy subjects. They also reported that rsfMRI complexity correlates with fluid intelligence. This finding was in line with the hypothesis in [17] where a positive relationship between intelligence and the temporal complexity of fMRI was reported. The prefrontal cortex and inferior temporal lobes were amongst brain regions with the strongest relationship between high fMRI complexity and high intelligence. These studies, to name but a few, suggest that the evaluation and analysis of fMRI fluctuations can provide insight into functional brain networks, the dynamics of brain structure and human behaviour. There is evidence that the temporal complexity of brain function supports different aspects of human behaviour and cognition [32]. Perturbed complexity across cortical areas may contribute to a range of brain diseases including epilepsy [33], Alzheimer’s disease [34], and schizophrenia [35]. Time-varying changes of functional brain networks are likely related to the fine balance between efficient information-processing and metabolic costs in the brain [36]. Spatial distribution of temporal complexity in brain function can shed light on how interactions between cortical regions are temporally organized and has the capacity of leading to imaging-based biomarkers of brain function in health and disease.

Different measures have been used for the temporal complexity analysis of fMRI including time-resolved graph theory measures [33], entropy measures [13,14,17,22,33], and self-similarity measures [21,30,37,38]. A crucial step for an appropriate interpretation of these measures is to develop relevant null distributions for scoring the grey boundary between complete randomness and pure regularity (i.e., temporal complexity) in fMRI [39,40]. An ideal null distribution must preserve all properties of the data except the one that is under investigation. Surrogate data analysis is a widely used data-driven approach for developing null distributions in real-world datasets. This approach is based on the shuffling of a single feature or a set of features in the data, while the other fundamental features are kept intact. In a complicated dynamical system with a combined complexity in time and space such as the human brain, appropriate null distributions must take both the dynamics and the underlying structure into account. Phase shuffling in the temporal Fourier domain is an effective way of generating the surrogates of fMRI when the cross-correlations between brain regions need to be preserved [41]. However, it fails to consider the anatomical basis of fMRI into account. Establishing a brain structure-informed statistical inference for temporal complexity analysis of fMRI is still an open question. A solution to this challenge is through graph signal processing [42] where fMRI time points are projected onto structurally informed basis vectors and randomization is performed in this joint domain [43].

In this study, we aim to perform an independent assessment on the most commonly reported aspects of temporal complexity in fMRI during rest and task engagement. We use two measures of temporal complexity, i.e., Hurst exponent and multiscale entropy, and compare them in the context of fMRI analysis. First, we validate the monofractal feature of brain hemodynamics during task engagement and rest using Hurst exponent in a population of unrelated subjects from the Human Connectome Project (HCP) [44]. Second, we assess task specificity of fMRI complexity during the task engagement and resting state. To this end, we perform a pair-wise support vector machine (SVM) analysis on the Hurst exponent and a multiscale entropy-based complexity index across all brain regions. Third, we examine the hypothesis of complexity alteration in brain hemodynamics due to task engagement. Fourth, we investigate the agreement between the spatial distribution of Hurst exponent and multiscale entropy in fMRI. Finally, we perform statistical testing on fMRI complexity through a brain structure-informed surrogate technique based on graph signal processing. We look into the mathematical properties of this technique in detail and its consequences for our analysis. Figure 1 illustrates the procedure of performing temporal complexity analysis and graph surrogate generation on a typical fMRI dataset, as adapted in this study.

## 2. Materials and Methods

### 2.1. Data and Preprocessing

We obtained the rest and task fMRI and diffusion-weighted scans of 100 unrelated subjects (ages 22–35) from the HCP1200 release [44]. Each subject underwent a number of fMRI recording sessions including four rsfMRI runs, seven task fMRI runs, and a diffusion MRI run. Each fMRI dataset had a voxel size of 2 × 2 × 2 millimetres and the repetition time ($T_{R}$) of 720 milliseconds in a 3-T Siemens Skyra scanner. We utilised two rest runs with left–right phase encoding as well as four task runs with a minimum length of 3 min or 250 $T_{R}$’s, i.e., Language, Motor, Social, and Working Memory. The other three task recordings, i.e., Gambling, Emotion, and Relational, were shorter than 3 min and therefore excluded from the analysis. Note that each task-based fMRI recording had a specific task design with a different number of conditions and trials (Table 1). See [45] for the description of the fMRI tasks. We included the rsfMRI datasets with left–right phase encoding only due to the known issue of asymmetric drop-out between left–right and right–left phase encoding of rest runs in the HCP database [46] and its potential impact on task specificity analysis in Section 2.3. Since the length of rsfMRI in all subjects was considerably longer than all task fMRI recordings (14.4 min versus 3 to 4 min), we used the first 399 $T_{R}$’s of rest runs in order to make the extracted complexity measures comparable. The fMRI datasets were preprocessed using SPM8 through a procedure described in [47]. First, fMRI datasets underwent a spatial smoothing by a 5 mm isotropic Gaussian kernel. Six motion parameters as well as average cerebrospinal fluid signal and white matter signal were then regressed out. We did not apply any further bandpass filtering on the fMRI time series at this stage. A parcellation mask [48] was used to parcellate the grey matter into 360 cortical regions of interest ($N_{ROI}$ = 360). The corresponding diffusion-weighted datasets were preprocessed through the steps outlined in [47] and used to extract the structural connectivity matrix of each subject for graph signal processing.

### 2.2. Temporal Complexity Analysis of fMRI

One of the most common ways to measure scale invariance in time series is by using the Hurst exponent *H* [49]. It determines whether there is a predominant time-scale or frequency component in the underlying dynamical process. The value of *H* varies between 0 and 1 where H≤ 0.5 implies short-memory or fast return to the mean (such as white noise), and H≥ 0.5 represents long-memory or a trending behaviour with random turning points. The value of *H* = 0.5 represents a random walk whose time points have no correlation with their past values. Scale-free signals have a long memory, because all of their time scales and spectral components contribute equally to their dynamics. This leads to a power law relationship in the spectral power of scale-free signals in the form of $P\left(f\right)\propto f^{-\beta }$ where P(f) is the power spectral density at frequency *f*, and β is a non-zero positive real number referred to as *spectral exponent*. For some scale-free processes such the fractional Brownian motion, there is a theoretical relationship between the spectral exponent β and the Hurst exponent *H* via the equation β=2H+1 [50]. In this study, we used detrended fluctuation analysis (DFA) [51] to estimate the Hurst exponent and log-linear line fitting to the power spectral density function to estimate the spectral exponent of fMRI time series. The DFA algorithm has been widely used in the past for the analysis of fMRI fractality [30,52]. Previous studies have reported an approximate range of 0.5–0.9 for the Hurst exponent of fMRI time series which highlights them as a class of signals with complex dynamic and long memory [30,52]. Given the direct link between the Hurst exponent and signal entropy [53], we also looked into the complex behaviour of fMRI using multiscale entropy [54]. This measure is based on the sample entropy [55] at several time scales of a signal x (here, mean fMRI time series at a particular ROI). Multiscale entropy characterises white noise by a very large entropy value (usually above 4) at the first time scale which rapidly decreases across coarser time scales. This leads to small areas under the multiscale entropy curve for highly irregular signals such as white noise. On the other hand, complex signals such as red/pink noise represent a relatively flat or monotonically increasing pattern across most of the time scales with a larger area under the curve compared to white noise [14]. Therefore, the area under the curve of multiscale entropy can be considered as a *complexity index* for temporal complexity analysis of biosignals [56]. See Appendix A for a detailed description of multiscale entropy.

### 2.3. Task Specificity of fMRI Complexity

In order to test the dependency of the complex properties of brain hemodynamics on task engagement, we used a set of SVMs with the radial basis function (RBF) kernels in order to evaluate the separability of the Hurst exponent and multiscale entropy over two rest runs and four task runs of fMRI. The SVM analysis is a supervised learning method which is widely used for performing classification and regression studies using fMRI datasets. We extracted ROI-wise complexity measures of $N_{subj}$ = 100 subjects and six fMRI runs. For each pair-wise comparison between the two fMRI runs, we considered each subject as an observation and the brain maps as feature vectors of size $N_{ROI}\times 1$ where $N_{ROI}$ = 360. This yielded 100 feature vectors of size $N_{ROI}\times 1$ for each fMRI run. Here, we investigated whether mental tasks can be discriminated in a population of healthy subjects using the ROI-wise temporal complexity of fMRI. We quantified the performance of SVM classifiers using the percentage of their classification loss or the proportion of observations misclassified by the model. This pair-wise SVM analysis led to two symmetric accuracy matrices of size 6×6 for the two temporal complexity measures. The SVM classifiers were evaluated via 5-fold cross-validation and their hyper-parameters were optimized through grid search. We minimized the issue of overfitting in the SVM classification analysis by optimizing the *box constraint* parameter in MATLAB’s *fitcsvm* function. This parameter controls the maximum number of support vectors, thereby preventing overfitting. The larger the box constraint parameter, the less support vectors are assigned to the SVM classifier.

### 2.4. Spatial Distribution of fMRI Complexity across Grey Matter

To be able to interpret the complexity measures of fMRI across brain regions, one must perform statistical testing. In this study, we generated $N_{Surr}$ = 100 surrogate fMRI datasets for each subject and each fMRI run through a *graph signal processing* framework which combines the brain structure and function in order to generate the surrogates of fMRI whose null distribution preserves the spatial smoothness of the fMRI signal on the structural connectome at each time point [42,43]. Our motivation for adapting this technique was to incorporate the underlying anatomical aspects of fMRI in the significance testing step, an important piece of information which is usually neglected in the fMRI complexity analysis studies. As discussed in Appendix B, the graph surrogate method: (*i*) preserves second order statistical moments across fMRI time points, i.e., temporal correlation; (*ii*) randomizes functional connectivity; (*iii*) randomizes the spatial variation of scale-free dynamics of fMRI at each single ROI; and (*iv*) randomizes the spatial variation of the scale-free dynamics between ROI pairs.

In order to obtain the group-level maps of brain complexity, we applied binomial testing on the subject-specific brain maps. First, each individual map of complexity was thresholded at a significance level of $\alpha_{subj}$ = 0.01 in order to obtain a binary map at the subject level. Then, the binomial distribution P(n) of having *n* detections was used at each ROI to examine the significant number of suprathreshold regions across subjects at the significance level of $\alpha_{group}$ = 0.001. It was equivalent with 10 detections (i.e., the number of suprathreshold regions) for a population of 100 subjects. The results were corrected for multiple comparisons at the number of regions and fMRI runs tested, i.e., 360 × 6 comparisons.

## 3. Results

### 3.1. FMRI Represents Complex Behaviour during Rest and Task

As seen in Figure 2, the normalized group mean power spectral density of all fMRI recording sessions (four fMRI runs and four task runs) show log-linearity in the frequency domain. This feature was more pronounced in rsfMRI datasets and the spectral exponents were RSN-specific with default mode, frontoparietal, and dorsal attention networks presenting the highest exponents in most cases (Figure 2B). The β exponents of different RSNs, however, were shown to be task-dependent. For example, the Social task led to the highest spectral exponent across subjects at the dorsal attention network or Rest runs led to the highest exponents at the default mode and frontoparietal networks. A striking observation was related to the existence of dominant peaks in the log-linear power spectral density functions of task fMRI in contrast to rsfMRI (Figure 2A). In order to rule out the influence of the task designs in this spectral feature of task fMRI recordings (Figure 2C), we regressed out the task timings trial-wise from the data and checked the spectral log-linearity of the residuals only. As Figure 2D illustrates, the peaks still remain in the log-linear power spectral density functions of task fMRI residuals, although they are slightly suppressed. This suggests that these spectral peaks are independent from the task design and likely explain why the Hurst exponent is generally smaller during task engagement and larger at rest [21,30,39].

### 3.2. Task Engagement Lowers Complexity of BOLD Activity

As summarized in the classification loss matrices of Figure 3, the fMRI of resting state and task engagement can be classified with high accuracy using both temporal complexity measures. However, the dynamics of fMRI sessions at rest are not distinguishable from each other (note the high classification loss between rsfMRI recordings in Figure 3C,D). Here, each dataset has been characterized as a set of $N_{subj}$ feature vectors of size $N_{ROI}\times 1$ where $N_{subj}$ = 100 and $N_{ROI}$ = 360. The distribution of temporal complexity across brain areas in different rest and task fMRI runs (brain maps of Figure 3A and Figure 4) suggests that the spatial profile of complex dynamics in fMRI is affected by task engagement and resting conditions. As evident in the histograms of individualized mean Hurst exponents across brain areas in Figure 3B, each task results in a regionally specific reduction in dynamic complexity, an observation in line with previous findings in the literature [30]. This establishes a framework for comparing the *task burden* in subjects. For example, the Hurst exponent histograms suggest that the working memory task is likely the most *demanding* brain state in the population of this study, because its associated histogram covers the lowest interval of self-similarity.

### 3.3. Complex Dynamics Exist in the Brain Structural-Functional Coupling

In Figure 5, we show the spectral power and multiscale entropy patterns of the projected fMRI datasets onto brain structure at rest and during task engagement. As Figure 5A illustrates, the distribution of spatial energy across the brain connectome follows a log-linear relationship. However, despite what we observed for the spectral power of fMRI in Figure 2A, a considerable difference between the power spectral distributions of graph signals extracted from different tasks and rest sessions is not evident here. This can also be observed in the multiscale entropy patterns and complexity indices of brain graph signals in Figure 5B,C which are quite comparable over four tasks and the average rest. In particular, there is an inflection point in the scatter plot of complexity indices (Figure 5C) associated with the colour transition in the multiscale entropy patterns of Figure 5B from dark blue (low randomness) to green (high randomness). These *elbow* or *knee* points are also indicative of log-linearity in the complexity domain of brain structure–function.

### 3.4. Spatial Patterns of Complex Dynamics in fMRI

Figure 6A illustrates the group-mean spatial patterns of multiscale entropy-based complexity index across brain areas for the average rest runs and four tasks. Furthermore, Figure 6B presents the pie chars of the percentage of suprathreshold ROIs in 7 RSNs associated with Figure 6A. All maps were thresholded at the subject-level *p*-value of 0.01 and family-wise error-corrected at the group-level *p*-value of 0.01 using the graph surrogate data generation method introduced in [43]. Furthermore, Table 2 summarizes the contribution of 7 RSNs in the suprathreshold brain regions of Figure 6. In all fMRI runs, the visual areas were amongst the regions with highest complex dynamics. The next mostly engaged RSNs in all runs were the dorsal attention network with a maximum cover of 7.5% followed by the frontoparietal and default mode networks with a maximum cover of 8.9%, all during the Working Memory task. Several regions across dorsal lateral prefrontal cortex (*DLPFC*) showed up as the most frequently observed areas with the highest complexity across all tasks and rest runs. These regions included *8Av*, *9m*, *8BL*, *10d*, *9a* and *p10p*. Other most repeated suprathreshold regions included: left *RSC* (anterior cingulate cortex), left *TE1a* (middle temporal gyrus), *PGi* (temporo–parieto occipital junction), *PGs* (inferior parietal cortex) and right *45* (inferior frontal cortex). The limbic network represented the least complex regions with most random behaviour in all task and rest runs. See [48] for the anatomical description of these brain labels.

## 4. Discussion

This study reinforces the existence of complex dynamics in brain function [22] and provides further evidence for the hypothesis of distinct complexity features in human behaviour and cognition [30]. Our results suggest that: (*i*) task-based and rsfMRI signals exhibit temporal complexity, inferred by Hurst exponent and multiscale entropy; (*ii*) rest and task periods of brain function can be distinguished from each other with high accuracy based on their temporal complexity profiles; (*iii*) cognitive load can suppress the complex dynamics of fMRI in contrast to the resting state; (*iv*) spatial distribution of Hurst exponent and entropy-based complexity index in fMRI are highly correlated; and (*v*) the visual, frontoparietal, default mode, and dorsal attention networks represent maximal complex behaviour compared to the rest of the brain in most mental states.

Temporal changes of neural activity in the brain are observed on the scale of milliseconds in single-cell spiking to the order of seconds in BOLD fluctuations [57]. In contrast to the traditional neuroscientific view which would mostly consider this variability as a random disturbance and measurement noise, many systematic patterns of information have been detected in neural variability over multiple temporal and spatial scales [54,58,59]. Neural variability at the BOLD level not only reflects inter-subject differences such as behavioural traits, but it also captures within-subject changes such as the dynamical states of the brain and cognitive load [57]. An important feature of neural variability in the brain is its temporally complex behaviour. Although temporal variability and temporal complexity are closely related, these two concepts are not necessarily the same. In other words, temporal complexity always exhibits variability, but a variable process is not necessarily complex. Also, high complexity is not necessarily equivant with high entropy. As a matter of fact, Gaussian white noise represents the highest signal entropy, highest variability, and highest unpredictability amongst all signal types, but it is not deemed as *temporally complex* according to our definition. A complex pattern of brain activity is rich in information over time and represents a balanced dynamic between order and disorder within brain networks or between different brain regions [13,14]. Recent theories in neuroscience have proposed that the temporal complexity of brain function is likely associated with information processing in the brain [13,59]. The first theory [60] indicates that a healthy brain retains an optimal degree of instability which allows it to enter to *different dynamical states* and sample the external world. In this way, the brain learns how to optimize responses to environmental stimuli. The second set of theories [61] argue that a moderate level of randomness is necessary in the neural system because it increases the probability of *neuronal firing* in subthreshold neurons. In contrast to the second proposal, a third view [62] is that the temporal complexity of brain function can enhance or suppress the likelihood of *neural synchrony* between cortical areas. All of these theories speak to the direct relationship between the temporal complexity of brain function and information transfer across neural populations, cortical regions and functional networks.

Considering the key role of complex processes in brain mechanisms, one would expect to find realizations of temporal complexity in brain hemodynamics as well. To investigate this possibility, it is crucial to quantify the temporal complexity of neural variability. Signal entropy measures were found to be relevant for investigating the complexity of brain function. This is mainly because these measures do not rely on distributional assumptions (unlike variance-based measures), and are not restricted by sinusoidal signal waveforms (unlike frequency-based measures). In this study, we utilized two measures for the temporal complexity analysis of fMRI in order to cross-check the fMRI complexity analysis results: multiscale entropy as an entropy-based measure which evaluates similar patterns of information throughout signals and the Hurst exponent as a variance-based measure which quantifies the memory of signals and their tendency to return to their mean values. Multiscale entropy has been shown to be sensitive to RSN-specific hemodynamics, and reproducible across healthy subjects [13,14]. The direct relationship between multiscale entropy and self-similarity [53] makes it a good candidate for investigating the properties of the Hurst exponent in fMRI.

It is important to note that the existence of self-similarity and fractality in BOLD signals has been subject to discussion in recent decades, mainly due to the limited temporal resolution of fMRI and sluggishness of the hemodynamic responses in the brain [63]. A fundamental question here is, even assuming the presence of temporal complexity in brain hemodynamics, that of whether the recorded BOLD signal inside an MRI scanner is able to adequately capture it. To address this question, the following issues should be considered: (*i*) low temporal resolution of the measured BOLD changes with a typical $T_{R}$ value in the scale of hundred milliseconds to seconds; (*ii*) the use of short-length fMRI time series (less than 200 $T_{R}$’s) in some studies; (*iii*) potential impact of preprocessing steps and residual scanning artifacts on the nonlinear dynamics of fMRI; (*iv*) upon the agreement on the complex dynamics of fMRI, whether it is monofractal or multifractal [21,64]. It has been shown that non-fractal time series with inadequate memory can still exhibit log-linear spectral power and be falsely identified as complex processes [63]. Therefore, one may argue that the observed complex behaviour of fMRI is not biological, but it simply originates from the *signal aspects* of fMRI such as its inadequate length in the previous studies. In fact, it has been shown that the accuracy of fMRI fractal analysis can be affected by the number of brain volumes [64]. As Figure 4 shows, a linear association across brain regions was evident between the Hurst exponent and the area under multiscale entropy curves of rsfMRI at the group level. Although the two measures operate at different ranges and utilize different methodologies to quantify temporal complexity, their spatial agreement across brain regions is relatively high (Pearson correlation above 0.8). However, the distinction between the dynamics of rsfMRI and task fMRI is more apparent in the Hurst exponent brain maps (Figure 3) in contrast to the entropy-based complexity index brain maps (Figure 4). It speaks to numerous monofractal analysis techniques which one can utilise to estimate the temporal complexity of fMRI, though each method may vary in its strengths, biases, and sensitivity to the biological changes of brain function. In spite of these technical differences, several simulated and experimental studies supported the hypothesis of a power law distribution for the fMRI power spectrum over the frequency band of 0.01 Hz to 0.1 Hz [21,52,65,66]. See [67] for a recent review on this topic.

The magnitude of *H* across brain areas in our study (Figure 3) is comparable with the previously reported findings showing a typical range of ≈0.5–1 for *H* [30,68,69]. The spatial extent of fMRI temporal complexity in our results is maximal across the frontoparietal, dorsal attention, visual and default mode networks and minimal across deep brain areas such as the limbic network (see Figure 3 and Figure 4). The considerable overlap between the analysis results of this study with the existing literature suggests that the fMRI signal length (minimum of 263 $T_{R}$’s—see Table 1) has been enough to replicate the previous hypotheses about the temporal complexity of brain function. In fact, a previous study has shown that valid Hurst exponents can be obtained from fMRI time series as brief as 40 s (≈56 $T_{R}$’s) through the DFA algorithm [30]. Furthermore, this implies that the fMRI preprocessing steps have not significantly manipulated the true dynamics of fMRI datasets in this study. According to our results, the spatial patterns of the Hurst exponent in fMRI are highly correlated with the associated patterns of entropy-based complexity index. This indicates that the two measures of fMRI temporal complexity converge to a similar outcome even though their computation is completely different.

The results of this study suggest that the temporal complexity of some regions such as motor areas is task-dependant. Higher values of complexity index and Hurst exponent of a given brain region or RSN (Figure 7) imply that the associated fMRI signals are temporally redundant and more predictable. Regions/networks with slower dynamics are likely responsible for the process of internal stimuli with low surprise and high adaptability. However, external stimuli such as sensory inputs and auditory inputs may reduce the adaptability of brain dynamics, increase surprise and shift the temporal complexity of brain function towards faster dynamics. An exception would be the visual network which shows variable dynamics from slow to fast across different tasks (see Figure 7), likely due to the larger capacity of visual areas in contrast to the other networks and brain regions. It is important to note that despite the presence of complex dynamics in fMRI, the relationship between fMRI temporal complexity and intrinsic functional connectivity is scale-dependant. It has been shown that the weighted sum of functional links to a given brain node, referred to as *functional connectivity strength* or FCS, is associated with the functional significance of that node in support of the information transfer across the brain [70]. The link between the FCS and temporal complexity of rsfMRI has been shown to be scale-dependant [13,14]. In particular, an inverse relationship has been reported between the FCS and temporal complexity of RSNs at fine time scales of multiscale entropy (τ≤ 5 at a $T_{R}$ of 0.72 s equivalent to time periods shorter than 3.5–4 s), while it turns to a proportional relationship at coarse time scales (τ> 6 or time periods greater than ≈4 s). This scale-dependant relationship varies for different RSNs. For example, frontoparietal and default mode networks represent the highest correlation between resting state FCS and temporal complexity at fine scales and lowest correlation at coarse scales, while it is the opposite for somatomotor, sub-cortical and visual networks [13,14]. On the other hand, the fine time scales of fMRI were associated with the dynamics of local neural populations, whilst the coarse time scales are likely related to long-range functional connections [13]. Altogether, these observations would suggest that, in order to obtain a comprehensive picture about complex dynamics of fMRI, one must consider the anatomical locations (i.e., spatial distribution) of brain regions. This speaks to the necessity of appropriate spatiotemporal methods for the significance testing of fMRI temporal complexity which can account for brain structure and function at the same time.

Surrogate data testing is a powerful method for characterizing the statistical properties of time series. In this approach, we want to compare the measure of interest extracted from the original data, i.e., the alternative hypothesis $H_{1}$, to the distribution of the same measure obtained from a large number of surrogate data, i.e., the null hypothesis $H_{0}$. In the context of fMRI temporal complexity, the null hypothesis could be that fMRI time series at different brain regions are generated by some non-complex processes and their functional relationships are also non-complex. If the complexity indices of fMRI signals fall within the null distribution $H_{0}$, this means that these indices only rely on the statistical properties preserved under the null $H_{0}$. Otherwise, we can reject the null hypothesis and interpret fMRI temporal complexity as revealing statistical properties beyond $H_{0}$. A critical question here is how to specify the null hypothesis of scale-free dynamics in fMRI and how to remove this signal feature of interest from the original data in order to generate surrogates [71,72]. In this study, we chose the brain graph randomization technique [43] which shuffles the power spectral density of fMRI in the graph domain for each time point, while keeping its underlying anatomical properties. As analytically discussed in the Appendix B, the surrogate data in this study preserve the pair-wise correlation between fMRI time series between brain regions. However, the functional connectivity, spatial variation of complex dynamics and spatial variation of the complex dynamics of functional connectivity are randomized in the surrogate time series. Therefore, the complex dynamics of brain ROIs with suprathreshold complexity indices is significantly different from the chance level and likely play a key role in moderating the information transfer across the whole brain. According to Table 2 as well as Section 3.4, the supporting role of brain regions is maximal within the frontoparietal, dorsal attention, visual, and default mode networks such as *DLPFC* and *PGi*, regardless of the resting state or task engagement. This finding is in line with the previous studies showing the dominant role of these brain areas in internal processing and the facilitation of information exchange in functional brain networks [73,74].

A number of caveats need to be considered in interpreting the results of this study. From the technical perspective, one should be aware of the limitation of Hurst exponent and multiscale entropy in capturing multivariate relations between fMRI time series at multiple ROIs. Both complexity measures used in this study treat ROI-wise fMRI signals as a set of individual and independent time series. However, mean fMRI time series at different ROIs are often statistically dependent and correlated due to the smearing effect of hemodynamic changes in the brain and a possible effect of fMRI preprocessing steps such as spatial smoothing. Therefore, it is plausible to use the multivariate versions of temporal complexity measures for fMRI data analysis [75,76]. A systematic comparison between the univariate and multivariate measures of complex dynamics and temporal complexity remains for our future work. Another consideration should be given to the nature of BOLD signals as an indirect measure of neural activity and the reduced amount of information it carries, in contrast to other direct measurements with higher temporal resolution such as local field potentials. Although fMRI has a greater capacity to cover larger brain areas than the localized measurements of neural activity and can provide large-scale information about the complex properties of brain dynamics, its temporal complexity must be treated as an indirect property of brain function. In this study, we assumed monofractality in fMRI and compared the entropy-based complexity index of fMRI with the classical Hurst exponent at different ROIs. It would be informative to check the possible links between the multiscale entropy and multifractality of fMRI using longer fMRI datasets and higher temporal resolutions during rest and task engagement. We also focused on the complexity of fMRI and its spatial distributions in the time domain only. However, this is only one dimension along which the complexity of brain dynamics can be measured. A more holistic view would be to consider the complexity of brain function in both time and space and adapt multivariate measures for the complexity analysis of fMRI time series [77].

## 5. Conclusions

Temporal complexity is a reproducible aspect of fMRI during rest and task engagement. This feature of brain function is task specific and can be suppressed by cognitive load. FMRI complexity is a discriminative feature between rest and task in the brain functional domain, but not in the brain structural domain. A brain structure-informed statistical testing of fMRI complexity reveals several areas with suprathreshold temporal complexity within the frontoparietal, visual, dorsal attention, and default mode networks.

## Figures and Tables

**Figure 1 entropy-24-01148-f001:**
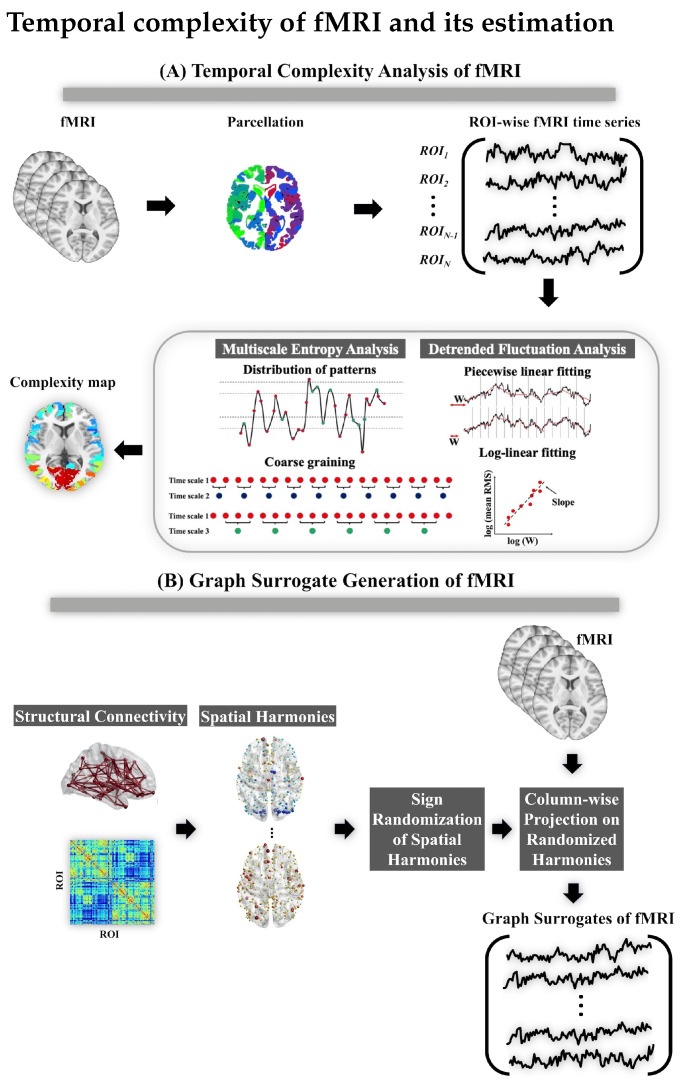
(**A**) The temporal complexity analysis procedure of fMRI in this study. (**B**) The process of generating graph surrogates from functional and structural MRI.

**Figure 2 entropy-24-01148-f002:**
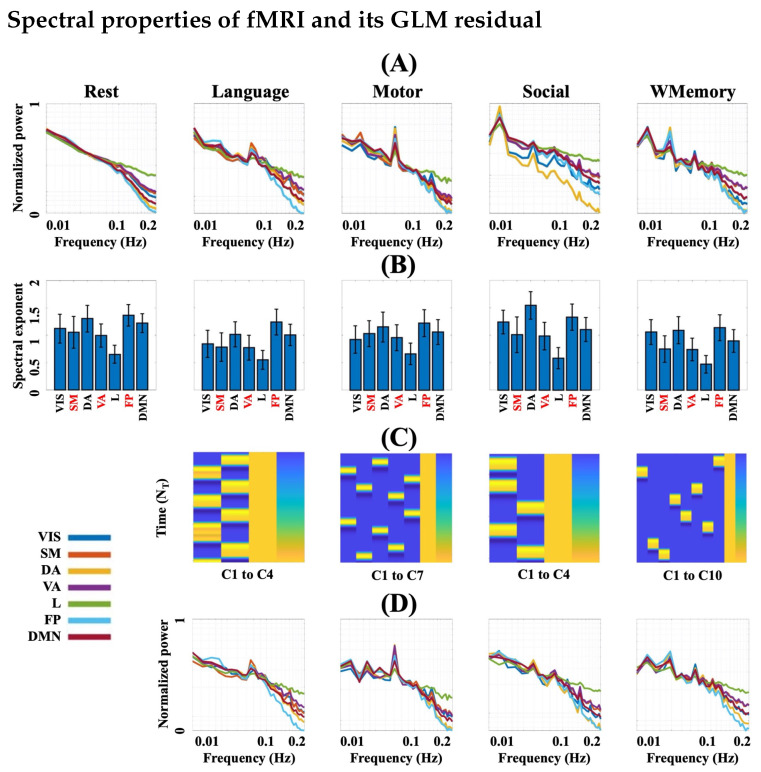
(**A**) Normalized power spectra of RSNs averaged over all subjects. (**B**) Corresponding β exponents as the slope of RSN-wise normalized logarithmic power spectra, estimated within the frequency band of 0.01–0.2 Hz. (**C**) Task fMRI protocol overview for Language, Motor, Social, and Working Memory tasks in HCP. Each yellow block represents an *event trial* and the trial blocks of each column in the event designs are identical. Each column represents a stimulus type referred to as a *condition* and has been denoted as $C_{i}$ in the figure. See [45] for the description of each condition in four HCP tasks. (**D**) RSN-wise normalized logarithmic power spectra averaged over all subjects, after regressing out the block designs from task fMRI through GLM. Abbreviations: GLM = general linear modelling; VIS = visual, SM = somatomotor; DA = dorsal attention; VA = ventral attention; L = limbic; FP = frontoparietal; DMN = default mode network; numbered C = Condition.

**Figure 3 entropy-24-01148-f003:**
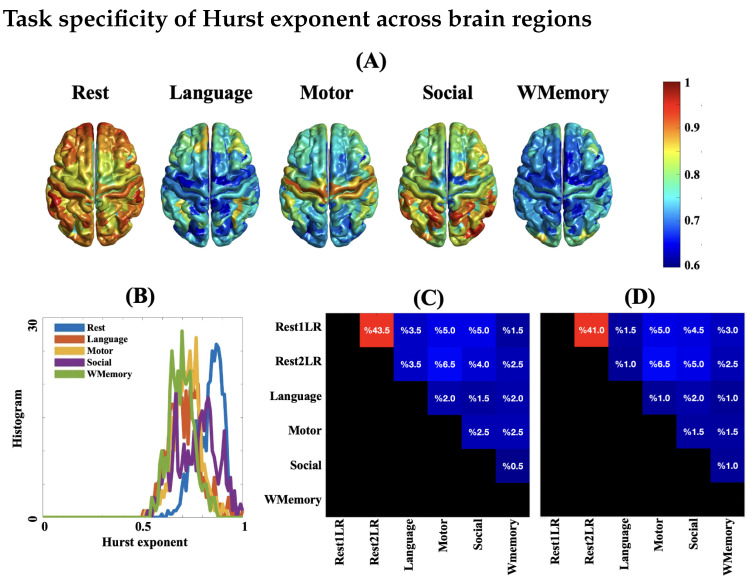
(**A**) Spatial distributions of the Hurst exponent across brain regions (averaged over subjects). The brain maps of 2 rest runs have been averaged. (**B**) Histograms of the group mean Hurst exponent over 360 brain regions for 4 task runs and 2 rest runs (averaged). Classification loss of pair-wise comparison of mental tasks using binary SVM classifiers with linear kernel: (**C**) Hurst exponent; and (**D**) multiscale entropy-based complexity index. The classification loss values have been color coded from dark blue (near zero) to bright red (near 1), and also mentioned on each pair. Abbreviations: WMemory = working memory; Rest1LR = first rest run with left-to-right slicing; Rest2LR = second rest run with left-to-right slicing.

**Figure 4 entropy-24-01148-f004:**
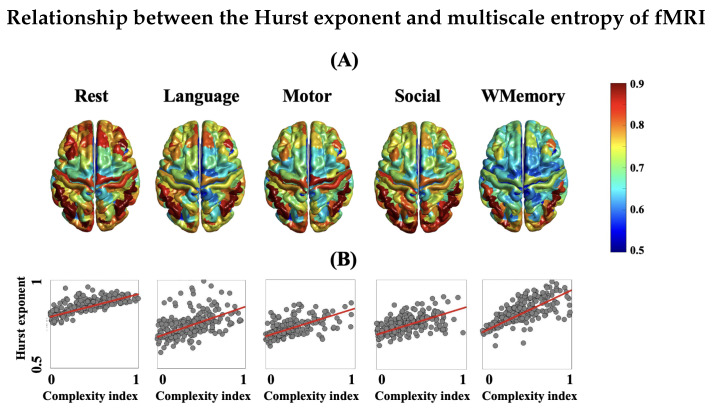
(**A**) Spatial distributions of the entropy-based complexity index across brain regions (averaged over subjects); (**B**) Joint distribution of the Hurst exponent and complexity index extracted from the rest and task fMRI datasets, averaged across all subjects. The brain maps of 2 rest runs are averaged. Abbreviation: WMemory = working memory.

**Figure 5 entropy-24-01148-f005:**
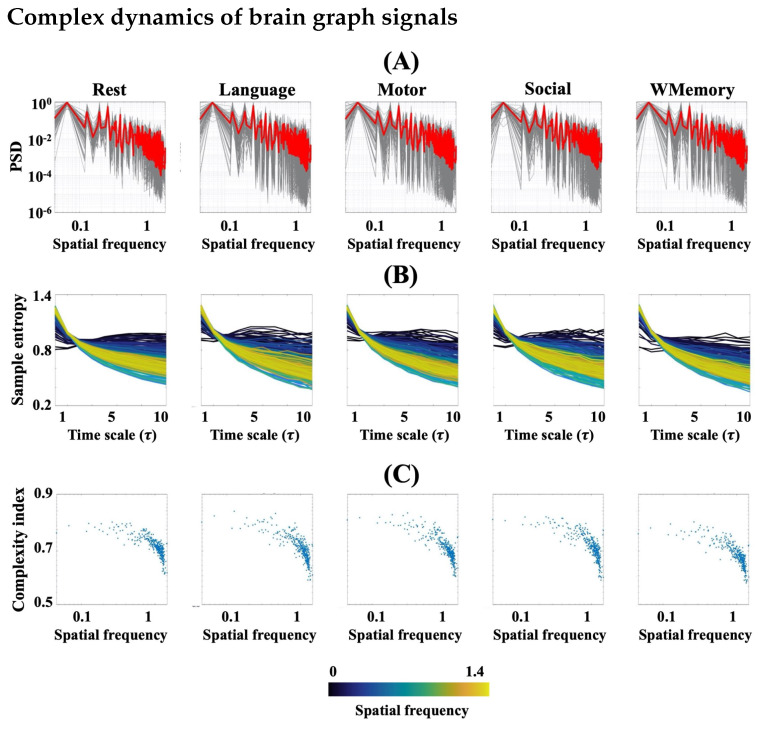
(**A**) Logarithmic plots of the power spectral density functions of brain graph signals (i.e., the projection of the fMRI data at rest and task onto brain structure) versus brain spatial harmonics. The plots of two rest runs are averaged. Each grey curve belongs to a single subject and the red curves represent group mean. All curves are normalized to 1. (**B**) Multiscale entropy patterns of the graph signals, colour coded by their associated brain spatial harmonyic (**C**) The complexity indices associated with the multiscale entropy curves of (**B**).

**Figure 6 entropy-24-01148-f006:**
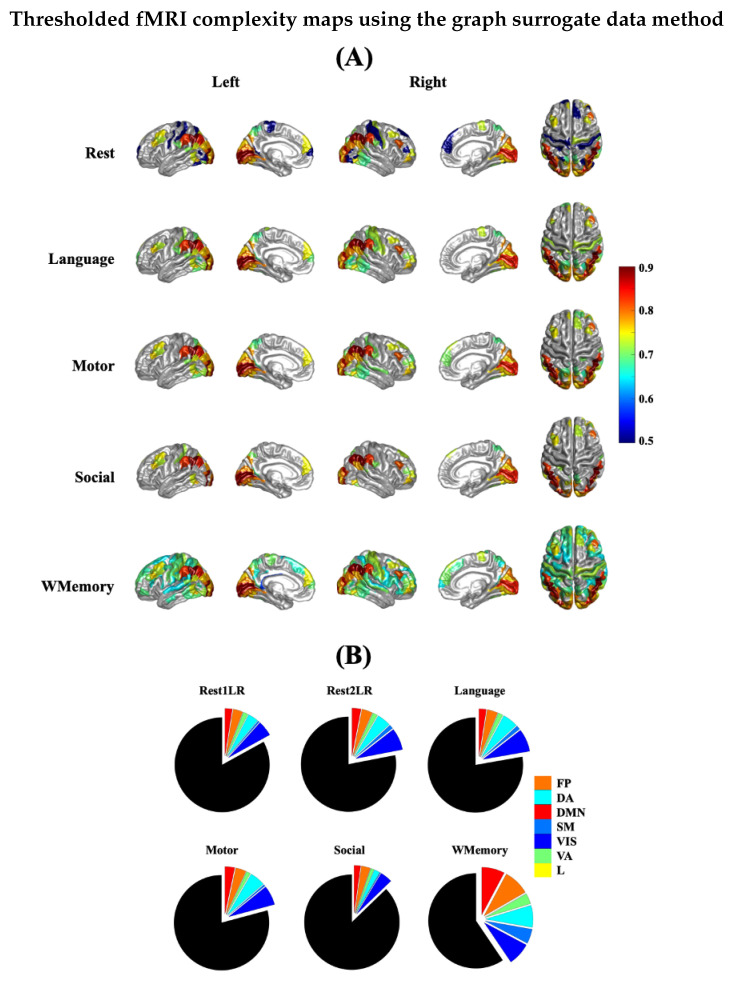
(**A**) Spatial distribution of group-mean fMRI temporal complexity across brain areas for 4 task runs and the average rest run. All maps are thresholded using the graph surrogate data generation [43] at the subject level *p*-value of 0.01 and family-wise error corrected at the *p*-value of 0.01. (**B**) Pie charts are the percentage of suprathreshold ROIs in 7 RSNs after graph surrogate testing, normalized by the number of ROIs. See Table 2 for the values of pie slices. Abbreviations: VIS = visual; SM = somatomotor; DA = dorsal attention; VA = ventral attention; L = limbic, FP = frontoparietal; DMN = default mode network; WMemory = working memory.

**Figure 7 entropy-24-01148-f007:**
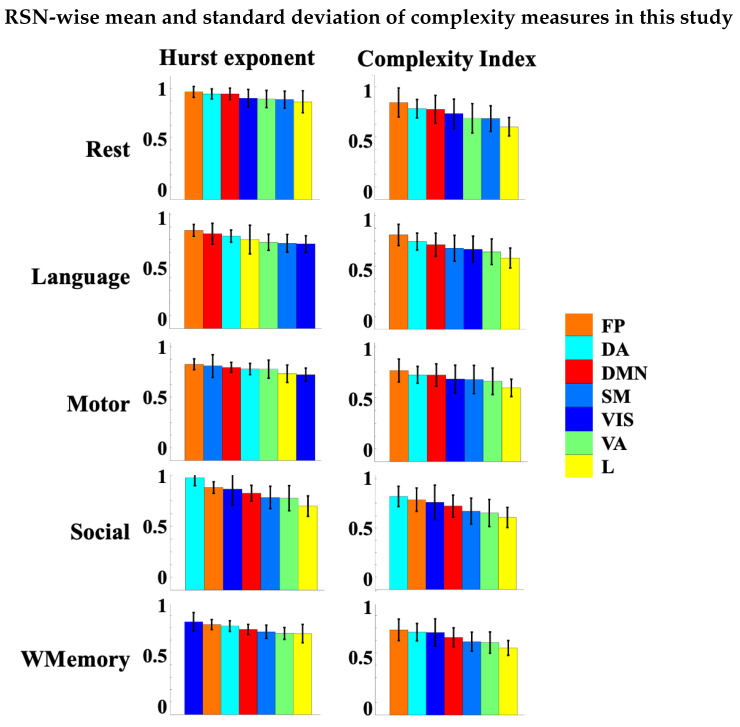
Group-level mean and standard deviation of the Hurst exponent and area under the curve of multiscale entropy at 7 resting state networks. Abbreviations: VIS = visual; SM = somatomotor; DA = dorsal attention; VA = ventral attention; L = limbic; FP = frontoparietal; DMN = default mode network; WMemory = working memory.

**Table 1 entropy-24-01148-t001:** List of fMRI runs of HCP, utilized in this study.

Run	Session	$N_{TR}$	Length in Minutes	No. of Conditions	No. of Trials
1	Rest1LR	399	4.8	-	-
2	Rest2LR	399	4.8	-	-
5	Language	305	3.67	2	11
6	Motor	273	3.29	5	10
7	Social	263	3.17	2	5
8	Working Memory	395	4.74	8	8

**Table 2 entropy-24-01148-t002:** Percentage of suprathreshold ROIs in 7 RSNs after the graph surrogate testing of brain complexity maps in Figure 6 (normalized by the number of ROIs). Abbreviations: VIS = visual; SM = somatomotor; DA = dorsal attention; VA = ventral attention; L = limbic; FP = frontoparietal; DMN = default mode network.

Task Name	VIS	SM	DA	VA	L	FP	DMN
Rest1LR	5.3%	0.6%	3.6%	1.7%	0%	3.3%	2.5%
Rest2LR	7.5%	1.4%	4.7%	1.7%	0%	3.6%	3.1%
Language	7.8%	1.4%	5.3%	1.7%	0%	3.6%	2.5%
Motor	6.7%	0.6%	5.3%	1.4%	0%	3.6%	3.3%
Social	4.2%	0.3%	1.9%	1.1%	0%	3.1%	2.2%
Working Memory	7.8%	5%	7.5%	3.6%	0%	8.9%	7.8%

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
