# Peer review of "On the Spatial Distribution of Temporal Complexity in Resting State and Task Functional MRI"

_entropy, 2022, doi:10.3390/e24081148_

Round 1

Reviewer 1 Report

I enjoyed reading this fine study with thorough design and methodologies, it adds to the growing literature on complexity of fMRI as an index of brain cognition. Only have a few minor suggestions

1) p3 2.1 In this study, we obtained the rest and task fMRI ...

2) The authors only used L-R phase encoding data instead of both directions for distortion correction using TOPUP, this may affect the accuracy of normalization and applying ROI template.

3) Appendix MSE calculation, the authors defined AUC MSE up to scale 10, this requires justification and/or reference. 

Reviewer 2 Report

Included in the attached file

Round 2

Reviewer 2 Report

I have no further suggestions and comments